# Improving Language Models' Meaning Understanding and Consistency by Learning Conceptual Roles from Dictionary

**Myeongjun Erik Jang**[1]    **Thomas Lukasiewicz**[2,1]

[1] Department of Computer Science, University of Oxford, UK
[2] Institute of Logic and Computation, Vienna University of Technology, Austria
myeongjun.jang@cs.ox.ac.uk, thomas.lukasiewicz@tuwien.ac.at

## Abstract

The non-humanlike behaviour of contemporary pre-trained language models (PLMs) is a leading cause undermining their trustworthiness. A striking phenomenon of such faulty behaviours is the generation of inconsistent predictions, which produces logically contradictory results, such as generating different predictions for texts delivering the same meaning or violating logical properties. Previous studies exploited data augmentation or implemented specialised loss functions to alleviate the issue. However, their usage is limited, because they consume expensive training resources for large-sized PLMs and can only handle a certain consistency type. To this end, we propose a practical approach that alleviates the inconsistent behaviour issue by fundamentally improving PLMs' meaning awareness. Based on the conceptual role theory, our method allows PLMs to capture accurate meaning by learning precise interrelationships between concepts from *word-definition* pairs in a dictionary. Next, we propose an efficient parameter integration technique that updates only a few additional parameters to combine the learned interrelationship with PLMs' pre-trained knowledge. Our experimental results reveal that the approach can concurrently improve multiple types of consistency, enables efficient knowledge integration, and easily applies to other languages.

## 1 Introduction

AI systems that behave like humans can be more reliable and trustworthy (De Visser et al., 2016; Jung et al., 2019). However, despite striking advances in various language tasks that modern PLMs have made, recent evidence of their non-humanlike behaviours (Hossain et al., 2020; Kassner and Schütze, 2020; Hosseini et al., 2021; Gupta et al., 2021; Sinha et al., 2021b) casts doubts on their trustworthiness, suggesting that PLMs' language understanding ability is far below that of humans.

The suggestive evidence revealing PLMs' incapability of meaning-understanding is *inconsistent* behaviours (Mitchell et al., 2022). Unlike humans, they exhibit logically contradictory behaviours in many respects (Jang et al., 2022a), such as generating different predictions for semantically identical texts (Ravichander et al., 2020; Elazar et al., 2021) or violating logical properties, e.g., the logical negation property (Kassner and Schütze, 2020; Ettinger, 2020; Asai and Hajishirzi, 2020; Jang et al., 2022b), symmetry (Wang et al., 2019c; Li et al., 2019; Kumar and Joshi, 2022), or transitivity (Asai and Hajishirzi, 2020; Lin and Ng, 2022). Such inconsistent behaviours undermine models' trustworthiness and limit their practical usefulness, particularly in risk-sensitive fields where even a minor undesirable action could lead to catastrophic consequences.

Previous works tried to alleviate the inconsistency issues through data augmentation (Ray et al., 2019) or introducing specialised loss functions (Elazar et al., 2021; Kim et al., 2021). These approaches, however, have limitations, as they can only address a specific type of consistency and are unable to handle others. For instance, consistency regularisation loss, which enforces a model's predictive distribution on the original and its paraphrased inputs to be similar (Elazar et al., 2021), cannot resolve the violation of the logical negation property or symmetry. Moreover, these approaches require expensive resources, which is practically inefficient, i.e., using a lot of computing resources for training large PLMs is impractical, and collecting auxiliary data (e.g., paraphrased texts) is demanding for low-resource languages.

To this end, we propose a comprehensive solution that can concurrently enhance multiple types of consistency in a resource-efficient manner. Based on previous studies that demonstrated the limitation of the distributional hypothesis (a fundamental theory underlying modern PLMs) in learning the

meaning of natural language (Sinha et al., 2021b; Jang et al., 2022b), we hypothesise that a leading cause contributing to the inconsistent behaviour is the incapability of PLMs to capture the precise meaning of language. Thus, our proposed approach is centred around mitigating PLMs' deficiencies by leveraging an auxiliary model that is trained to imitate human cognition, thereby enhancing the awareness of semantic meaning. Our underpinning assumption is the conceptual role theory, a convincing theory that accounts for human cognition. Based on the theory, by which the interrelationship between concepts predominantly determines a word's meaning (Deacon, 1998; Santoro et al., 2021; Piantadosi and Hill, 2022), our approach first learns abundant symbol meanings by tracking more precise interconnections through word-definition pairs in a dictionary. Next, we propose a training-efficient parameter integration method that combines the learned parameters with those of existing PLMs. This enables the fine-tuning process to update only a small number of parameters. Our contributions can be briefly summarised as follows:

1. We verify that enhancing PLMs' meaning awareness can improve multiple types of consistency concurrently.

2. We propose a training-efficient parameter integration, which allows practical fine-tuning for large-sized PLMs.

3. The proposed approach is readily applicable to low-resource languages.

4. The results suggest that the proposed approach allows further consistency improvements in a simple but effective way by aggregating the parameters of PLMs possessing different inductive biases.

## 2 Methodology

### 2.1 Learning Conceptual Roles from Dictionary

In an effort to imitate the human language understanding process, we employed the conceptual role theory, a convincing theory assuming that it is a concept's role in some greater mental theory that primarily determines its meaning (Piantadosi and Hill, 2022), where the key is the interrelation between concepts (Deacon, 1998; Santoro et al., 2021). For example, the meaning of "water" can be defined by other interlinked concepts like "liquid", "without smell", "hydrogen", and "oxygen".

The recent success of large-size PLMs supports the conceptual role theory (Piantadosi and Hill, 2022). PLMs are the contemporary NLP techniques that exploit the distributional hypothesis as their learning objective, i.e., masked language modelling (Sinha et al., 2021a)[1]. This hypothesis assumes that semantically analogous words will appear in similar contexts (Harris, 1954). Hence, PLMs learn to define the words' meaning through other words that co-occur in similar contexts. That is, they capture the interrelationship of concepts frequently appearing in analogous contexts. Although the distributional hypothesis is an efficient assumption allowing self-supervised training, it cannot perfectly capture the interrelation between concepts because the concepts can deliver different meanings even though they often appear in similar contexts. For example, it is difficult to capture semantic antonymy through the distributional hypothesis (Jang et al., 2022b).

To this end, we aimed to improve PLMs' understanding of meaning by making them learn more precise interrelationships. Recent studies revealed that the training data play a critical role in deciding the inductive bias rather than the model's structure or learning objective (Furrer et al., 2020; Wang et al., 2022; Jang et al., 2022a). Therefore, we designed a learning task that provides training instances where a concept and other closely interconnected concepts are presented together to a model having a language modelling objective. To achieve this, we used *word-definition* pairs from a dictionary, because a definition is a composition of words explaining the target word, and thereby, it has been the most commonly used and very effective tool for vocabulary learning, especially for second and foreign language learners (Takahashi, 2012; Zhang et al., 2020). A target word and its definitions were concatenated as a single text and used as a training instance for language modelling, allowing a model to determine a word's meaning based on highly related concepts rather than those that appear in similar contexts. For example, when it comes to the words "happy" and "unhappy", distributional models normally generate high similarity, because they are both emotional expressions and appear in similar contexts frequently [2]. However, our proposed task enables capturing precise interconnection between concepts: "unhappy" with "not happy", "sad", and "not pleased". An exam-

---

[1]See Appendix A for details.

[2]For instance, Skip-gram vectors (Mikolov et al., 2013) trained with English Wikipedia: [link].

ple of our training data is provided in Table 6 in Appendix C.

We introduced the intermediate training technique (Phang et al., 2018; Wang et al., 2019a; Liu et al., 2019a; Pruksachatkun et al., 2020; Vu et al., 2020) that first trains PLMs on an intermediate task and uses it for other downstream tasks. Hence, we retrained PLMs on our new dataset and named it conceptual role model (CRM). A leading cause for using the intermediate training is that the number of *word-definition* pairs was not sufficient enough to train large PLMs from scratch, and their weights can be used as good initial values. The CRM has practical advantages in that (1) it can be applied to any PLM trained with language modelling objectives, and (2) it is readily applicable to other languages due to the relative ease of collecting dictionary data.

## 2.2 Training-efficient Parameter Integration

We propose a parameter integration method to effectively incorporate the previous knowledge obtained by PLM with enhanced meaning awareness of CRM.

### 2.2.1 Problem Statement

Let $W_p \in \mathcal{R}^{d \times l}$ and $W_c \in \mathcal{R}^{d \times l}$ be the parameter matrices of the PLM and CRM, respectively. Note that the PLM and CRM share the same model architecture, i.e., having a parameter matrix of the same size. We aim to learn $W_{new} \in \mathcal{R}^{d \times l}$, i.e., an integrated parameter matrix, from $W_p$ and $W_c$ during fine-tuning. Thus, the process can be defined as follows:

$$W_{new} = W_o \begin{bmatrix} W_p \\ W_c \end{bmatrix}, \qquad (1)$$

where $W_o \in \mathcal{R}^{d \times 2d}$ is a learnable parameter matrix while $W_p$ and $W_c$ remain fixed during fine-tuning. By decomposing $W_o$ into $W_1 \in \mathcal{R}^{d \times d}$ and $W_2 \in \mathcal{R}^{d \times d}$, Eq. 1 can be rewritten as follows:

$$\begin{aligned} W_{new} &= \begin{bmatrix} W_1 & W_2 \end{bmatrix} \begin{bmatrix} W_p \\ W_c \end{bmatrix} \\ &= W_1 W_p + W_2 W_c \\ &= W'_p + W'_c, \end{aligned} \qquad (2)$$

where $W'_p$ and $W'_c$ denote the matrices after fine-tuning. Finally, decomposing $W'$ with a fixed (W) and updated part ($\Delta W$) allows us to reformulate Eq. 2 as follows:

$$\begin{aligned} W_{new} &= (W_p + \Delta W_p) + (W_c + \Delta W_c) \\ &= W_p + W_c + \Delta W_t. \end{aligned} \qquad (3)$$

As a result, $W_{new}$ becomes the addition of fixed matrices $W_p$, $W_c$, and a learned matrix $\Delta W_t \in \mathcal{R}^{d \times l}$.

Updating the whole parameters of $\Delta W_t$ is exactly the same as the fine-tuning with the pre-trained weights of $W_p + W_c$, which is impractical particularly for large-sized PLMs. Also, fine-tuning contains the risk of catastrophic forgetting, as studied in many previous works (Pruksachatkun et al., 2020; Wallat et al., 2020). To compensate for this, we introduce a low-rank adaptation technique (Hu et al., 2022) for the parameter integration, which fixes the pre-trained weights and updates only a few number of parameters based on PLMs' intrinsic dimension (Aghajanyan et al., 2021). Specifically, we transform $\Delta W_t$ to a multiplication of two matrices $A \in \mathcal{R}^{d \times r}$ and $B \in \mathcal{R}^{r \times l}$, where $r \ll \min(d, l)$:

$$W_{new} = W_p + W_c + AB. \qquad (4)$$

As a consequence, the number of trainable parameters is considerably reduced, enabling efficient fine-tuning for large-sized PLMs. Also, compared to other adapter modules (Houlsby et al., 2019; Pfeiffer et al., 2021) that introduce additional adapter modules between layers, the approach only adds up $AB$ without increasing the number of parameters. As a result, it avoids the use of any additional time or resources during the inference phase, which enables practical inference (Hu et al., 2022).

**Aggregating $W_p$ and $W_c$.** The addition of $W_p$ and $W_c$ in Eq. 3 causes the amplification of the weight scale, which prevents us from using or searching training hyperparameters based on values used in prior studies. Thus, we used the simple averaging aggregation method (Wortsman et al., 2022):

$$W_{new} = \frac{W_p + W_c}{2} + A'B'. \qquad (5)$$

The reason behind leveraging the simple averaging technique is to show the efficacy of CRM, i.e., achieving remarkable improvements in consistency through the most straightforward aggregation method. However, alternative aggregation methods, such as Fisher-Weighted Average (Matena and Raffel, 2022) or RegMean (Jin et al., 2023), can also be employed.

**Additional Knowledge Add-up.** Note that the aggregation further allows us to integrate the

weights of any other models if they share the same structure with the PLM and CRM. Thus, Eq. 5 can be rewritten as follows:

$$\mathrm{W}_{new} = \frac{1}{|S| + 2} \sum_{i \in \{p,c,S\}} \mathrm{W}_i + \mathrm{A}'\mathrm{B}', \quad (6)$$

where $S$ is the set of additional PLMs' weights that are going to be added. Note that, as the parameters are integrated through an addition, no additional training/inference resources are required during fine-tuning, which is computationally beneficial.

**Applying the approach to PLMs.** As modern PLMs have a transformer (Vaswani et al., 2017) as a backbone architecture, we followed Hu et al. (2022) to apply the low-rank adaptation technique, i.e., limiting its usage to self-attention weights and excluding MLP weights from the scope. We also used the same hyperparameters found by Hu et al. (2022), i.e., for a hidden representation $x$, $\Delta \mathrm{W}'_t x$ is scaled by $\frac{\alpha}{r}$, where $\alpha$ and $r$ are set to 16 and 8, respectively.

## 3 Experiments

### 3.1 Training the CRM model

**Dataset.** To collect *word-definition* pairs, we first collected English words from the List of English Words [3] and Wikipedia Word Frequency data [4], and extracted word meanings through the Wikipedia English Dictionary API. As a result, 455K word-definition pairs were collected for training the CRM model.

**Model and training details.** We used RoBERTa (Liu et al., 2019b) as a backbone model for our experiments. Both base- and large-size models were used to observe the influence of the model's size. The base- and large-size models were trained for 500K and 700K steps, respectively. We used the AdamW optimiser (Loshchilov and Hutter, 2017) with a learning rate of 1$e$-6 and a linear learning rate scheduler decaying from 1$e$-2. The $\beta_1$, $\beta_2$, and $\epsilon$ values were set to 0.9, 0.98, and 1$e$-6, respectively. Four GeForce GTX TITAN XP GPUs were used for the training of the CRMs.

### 3.2 Evaluation on the English Dataset

We evaluated the proposed approach on two existing benchmark datasets: GLUE (Wang et al., 2019b) for measuring basic NLU task performance and BECEL (Jang et al., 2022a) for evaluating the consistency.

### 3.2.1 Experiments on the GLUE Benchmark

We trained the proposed model to GLUE downstream tasks to ascertain whether the approach performs well on widely used NLU tasks. The models that only leverage PLM weights (i.e., zero weights for the CRM) were also trained for comparison. We repeated the experiments five times for tasks with a training set having more than 100K data points (e.g., MNLI, QNLI, QQP, and SST2) and ten times for the other tasks. Detailed training hyperparameters are presented in Appendix B. The evaluation metric is Matthew's correlation for COLA and accuracy for the others. The results of the validation set are summarised in Table 1. For RoBERTa-base, our approach performs better than using only the PLM weights in three tasks, i.e., COLA, MRPC, and RTE, and shows a comparable performance in other tasks. For RoBERTa-large, two approaches produce a similar performance in most tasks apart from COLA, where our approach performs significantly better. It is captivating that integrating CRM weights can significantly improve the performance of the COLA task, which is designed to check grammatical errors, i.e., requires linguistic knowledge. Also, the improvements are more significant in RoBERTa-base, where the PLM stores less pre-trained knowledge. The experimental results suggest that CRM weights provide more abundant linguistic knowledge, which can help enhance the understanding of text meaning.

### 3.2.2 Experiments on the BECEL Dataset

Next, we assessed our approach on the BECEL dataset that allows the evaluation of multiple consistency types over various downstream tasks. For each task, a model was fine-tuned based on the original training set and evaluated on the test sets, specially designed to measure various consistency types. We refer to the original paper for more detailed information. We did not consider the additive consistency in this work, because it was reported that most PLMs performed well in the additive consistency (Jang et al., 2022a). Two downstream tasks, SST and AG-News, were omitted from our evaluation scope, because they contain evaluation sets for semantic consistency only. To ascertain the influence of the CRM and parameter integration technique, we investigated four evaluation scenar-

[3][Link], accessed 20th January, 2023.
[4][Link], accessed 20th January, 2023.

| Model | | COLA | MRPC | RTE | QQP | SST2 | MNLI | QNLI | Avg |
|---|---|---|---|---|---|---|---|---|---|
| RoB$_B$ | P only | 62.2 | 89.1 | 77.1 | **89.8** | 94.2 | **87.0** | **92.6** | 84.5 |
| | P+C | **63.2*** | **89.5** | **79.1*** | **89.8** | **94.5** | 87.0 | 92.6 | **85.1** |
| RoB$_L$ | P only | 67.3 | **90.5** | 86.6 | 90.9 | 95.9 | **90.4** | **94.8** | 88.0 |
| | P+C | **68.4*** | **90.5** | **86.9** | **91.1†** | **96.0** | 90.4 | 94.8 | **88.3** |

Table 1: Average GLUE performance of RoBERTa-base (RoB$_B$) and RoBERTa-large (RoB$_L$) models when the parameter integration is applied. The best performance is in bold. The figures of the proposed approach (P+C) show a significant difference compared to our counterpart (P only) with $p$-value $< 0.05$ (*) and $p$-value $< 0.1$ (†) using the t-test.

| Model | | BoolQ | | MRPC | | | RTE | | | SNLI | | | | WiC | | |
|---|---|---|---|---|---|---|---|---|---|---|---|---|---|---|---|---|
| | | $\tau_{sem}$ $\tau_{neg}$ | | $\tau_{sem}$ $\tau_{neg}$ $\tau_{sym}$ | | | $\tau_{sem}$ $\tau_{neg}$ $\tau_{sym}$ | | | $\tau_{sem}$ $\tau_{neg}$ $\tau_{sym}$ $\tau_{trn}$ | | | | $\tau_{sem}$ $\tau_{sym}$ $\tau_{trn}$ | | |

| Model | | $\tau_{sem}$ | $\tau_{neg}$ | $\tau_{sem}$ | $\tau_{neg}$ | $\tau_{sym}$ | $\tau_{sem}$ | $\tau_{neg}$ | $\tau_{sym}$ | $\tau_{sem}$ | $\tau_{neg}$ | $\tau_{sym}$ | $\tau_{trn}$ | $\tau_{sem}$ | $\tau_{sym}$ | $\tau_{trn}$ |
|---|---|---|---|---|---|---|---|---|---|---|---|---|---|---|---|---|
| RoB$_B$ FT (125M) | P-only | 11.3 | 43.2 | 7.4 | 82.2 | **2.7** | 11.9 | 22.9 | 1.0 | 10.8 | 8.3 | 12.0 | 3.8 | 14.8 | **4.0** | 13.6 |
| | P+C | **10.7†** | **35.4*** | **6.7** | **78.9†** | **2.7** | **11.4** | **18.3†** | **0.5*** | **10.6** | **7.4*** | **11.2†** | **3.7** | **13.8** | 4.2 | **12.9** |
| RoB$_B$ PI (0.8M) | P-only | 12.2 | 59.9 | 10.0 | 82.2 | 6.5 | 12.8 | 30.9 | 1.4 | **9.8** | 8.3 | 11.8 | 4.1 | 15.1 | 3.5 | 13.9 |
| | P+C | **11.4*** | **50.7*** | **7.8*** | **71.1*** | 6.3 | **11.9†** | **26.4†** | 1.7 | 10.0 | **7.3*** | 12.1 | 4.0 | **11.9*** | 3.5 | **13.5** |
| RoB$_L$ FT (355M) | P-only | 8.4 | 45.9 | **6.2** | 78.9 | 2.6 | 7.8 | 7.0 | 2.1 | 9.2 | 7.8 | 9.2 | 3.3 | 10.6 | 5.9 | 11.5 |
| | C-only | 8.1 | 45.4 | **6.2** | **72.8†** | 2.2 | 7.2* | 7.7 | 1.4† | 9.2 | 6.0* | 9.1 | **3.2** | 11.0 | **4.7** | 10.8 |
| | P+C | **8.0†** | 45.8 | **6.2** | 75.2 | 2.0 | **6.2*** | 7.0 | 1.3† | **9.0** | 5.9* | 8.8 | 3.2 | **8.9†** | 4.8 | **9.6†** |
| RoB$_L$ PI (2.6M) | P-only | 9.6 | 43.3 | 6.8 | 77.3 | 4.7 | 7.7 | **7.0** | 2.2 | **8.6** | 7.3 | 7.3 | 3.2 | 11.7 | 3.0 | 10.6 |
| | C-only | **8.1*** | 41.9 | **6.4** | 74.8 | **2.0*** | 6.6† | 7.5 | 2.1 | 8.8 | 5.3* | 7.3 | 3.2 | 12.2 | 3.8 | **9.5†** |
| | P+C | 8.7* | **37.8*** | 6.7 | 77.0 | 4.4 | **6.5†** | 7.2 | **1.6†** | **8.6** | **5.2*** | 7.1 | **3.0*** | 10.9 | 2.8 | **9.5†** |

Table 2: Average BECEL performance of RoBERTa-base (RoB$_B$) and RoBERTa-large (RoB$_L$). $\tau_{sem}$, $\tau_{neg}$, $\tau_{sym}$, and $\tau_{trn}$ denote semantic, negational, symmetric, and transitive inconsistency, respectively. All evaluation metrics are lower the better. For each evaluation scenario, the best figure is highlighted in bold. The values of P+C and C-only show a significant difference compared to P-only with $p$-value $< 0.05$ (*) and $p$-value $< 0.1$ (†) using the t-test.

ios determined by (1) whether to only use the PLM (P-only) or exploit the CRM (P+C), and (2) fine-tuning the whole parameters (FT) or applying the parameter integration (PI). Regarding RoBERTa-large models, we additionally introduced models that exclusively employ CRM (C-only) for more detailed analysis. For clarification, P+C with FT denotes a model that fine-tunes the whole parameters where the initial weights are set to the aggregation of the PLM and CRM weights, and P-only with PI refers to a model that only introduces a low-rank adaptation technique to the PLM, i.e., the work of Hu et al. (2022). Detailed information regarding the training hyperparameters is provided in Appendix B. For the SNLI task, having a large training set, we repeated the experiments five times for each model and ten times for the other tasks. When it comes to the evaluation metrics, we basically followed the metrics used by Jang et al. (2022a) but made a marginal modification in negation and symmetric consistency metrics. The two consistency types are conditional and, therefore, apply to examples having specific labels. Jang et al. (2022a) used gold labels for assessing the condition, yet this approach entails a risk of misestimation, because it includes instances where the model makes incorrect decisions. Hence, to avoid the risk, we adopted a more conservative evaluation metric that exclusively considers instances in which the model makes accurate predictions, i.e., the metrics are calculated based on the model's belief. The average performance is provided in Table 2.

**Influence of the CRM.** Experimental results show that leveraging the CRM weights can improve multiple types of consistency across many downstream tasks. Specifically, among 15 consistency test cases, the models using both PLM and CRM exhibit lower inconsistencies than their counterparts using only PLM. RoBERTa-base models with the CRM weights exhibit statistically significant improvements in 7 cases for FT and PI. The figures for the RoBERTa-large models are 6 and 7, respectively. It is captivating that the improvements are very significant in consistency types requiring meaning-understanding, i.e., semantic and negational consistency. The experimental results support our hypothesis, stating that learning pre-

| Model | BoolQ | | | MRPC | | | | RTE | | | | SNLI | | | | | WIC | | | |
|---|---|---|---|---|---|---|---|---|---|---|---|---|---|---|---|---|---|---|---|---|
| | $\mathcal{A}$ | $\tau_{sem}$ | $\tau_{neg}$ | $\mathcal{A}$ | $\tau_{sem}$ | $\tau_{neg}$ | $\tau_{sym}$ | $\mathcal{A}$ | $\tau_{sem}$ | $\tau_{neg}$ | $\tau_{sym}$ | $\mathcal{A}$ | $\tau_{sem}$ | $\tau_{neg}$ | $\tau_{sym}$ | $\tau_{trn}$ | $\mathcal{A}$ | $\tau_{sem}$ | $\tau_{sym}$ | $\tau_{trn}$ |
| PI (P+C) | 85.6* | 8.7* | 37.8† | 90.5 | 6.7† | 77.0 | 4.4* | 86.9 | 6.5 | 7.2* | **1.6*** | **93.0*** | 8.6* | **5.2*** | **7.1*** | 3.0* | 73.0 | 10.9* | **2.8** | 9.5* |
| PI (P+C+M) | **85.7*** | 8.3 | **33.9†** | 90.3 | 6.3 | **69.3*** | 4.3* | **87.6*** | 6.5 | **5.3*** | 1.8* | **93.0*** | 8.3* | **5.2*** | 7.2* | 2.9* | **74.2*** | 8.6† | 3.0 | **8.9*** |
| DictBERT | 74.6 | 8.7 | 68.1 | 87.6 | 11.2 | 86.5 | 5.7 | 74.6 | 15.0 | 38.0 | 2.5 | 90.7 | 10.5 | 10.8 | 10.0 | 3.7 | 67.3 | 13.3 | 3.4 | 16.6 |
| Sem-Aug | 84.6 | 8.2 | 44.1 | 90.3 | **5.3** | 80.4 | **1.2** | 86.2 | **6.3** | 12.1 | 3.7 | 56.3 | 7.6 | 17.9 | 10.2 | 1.1 | 72.5 | 8.6 | 4.7 | 11.2 |
| Sem-CR | 84.3 | **7.8** | 67.6 | **90.7** | 5.8 | 78.7 | 2.7 | 83.3 | 7.1 | 13.8 | 2.8 | 55.5 | **6.6** | 16.9 | 11.8 | **1.3** | 69.9 | **6.6** | 3.8 | 11.6 |

Table 3: Performance comparison of our proposed approaches and baseline models. M denotes a RoBERTa-large model trained on the meaning-matching task. $\tau_{sem}$, $\tau_{neg}$, $\tau_{sym}$, and $\tau_{trn}$ denote semantic, negational, symmetric, and transitive inconsistency, respectively. $\mathcal{A}$ refers to the validation accuracy. The best figure is highlighted in bold. The values of the proposed approaches show a significant difference compared to the best-performing baseline with $p$-value $< 0.05$ (*) and $p$-value $< 0.1$ (†) using the t-test.

cise conceptual roles can improve PLMs' meaning awareness and, therefore, can improve consistencies. However, the improvements in consistency types requiring the understanding of logical relations, i.e., symmetric and transitive consistency, are relatively marginal, particularly in small-sized models. This indicates that enhancing meaning awareness has a limitation to improving logical consistencies, and another solution is required for further improvements. We leave this as future work.

**The effect of using both CRM and PLM.** In order to ascertain whether the combined use of both CRM and PLM offers greater benefits compared to utilising only CRM, we conducted training with RoBERTa-large models that exclusively rely on CRM for both FT and PI. First, we observe that C-only models achieve an enhanced consistency in general compared to P-only models, but the improvements are less significant compared to P+C models. Under $p$-value $< 0.1$, C-only models exhibit statistically significant improvements in 4 and 5 out of 15 test cases in FT and PI, respectively, while the figures of P+C models are 5 and 7, respectively. Also, it is confirmed that P+C models generally produce a significantly lower inconsistency than C-only models in several test cases under $p$-value $< 0.1$. These cases involve semantic and transitive consistency in the WiC task and negational consistency in the RTE task regarding FT. For PI, negational consistency in the BoolQ, semantic consistency in the WiC, transitive consistency in the SNLI task, and symmetric consistency in the RTE and WiC tasks correspond to these cases. Moreover, the inconsistency of C-only models is even higher than P-only models in some cases, while P+C models do not exhibit a statistically significant performance degradation. The experimental findings suggest that P+C models exhibit higher sta-

bility and achieve more significant improvements in consistency compared to the use of only CRM. This underscores the advantage of incorporating pre-trained knowledge from PLM and the enhanced meaning understanding of CRM.

**Fine-tuning vs. parameter integration.** The leading cause of introducing PI is to enhance the training/inference efficiency of large-size PLMs. Applying PI considerably decreases the number of training parameters, from 125M to 0.8M for RoBERTa-base and from 355M to 2.6M for RoBERTa-large, respectively. For RoBERTa-base, we observe several cases where introducing PI increases the inconsistency metrics in both P-only and P+C models, 4.5 over 15 test cases on average under $p$-value $< 0.1$. The aggravation mostly occurred in downstream tasks having a small training set, i.e., BoolQ, MRPC, and RTE, while the performance marginally changed in other tasks with a large training set. However, this is not a severe demerit, because fine-tuning the base-size model on a small training set is not a resource-consuming job. In contrast, when considering RoBERTa-large, the degradation in inconsistency metrics is observed in 2 out of 15 test cases on average, with a $p$-value $< 0.1$, which is considerably lower compared to RoBERTa-base. Additionally, the inconsistency is improved in 5.5 out of 15 test cases on average, which stands in contrast to RoBERTa-base, where no statistically significant improvements are observed with the use of PI. The results suggest that leveraging PI is more beneficial to large-size models, because it not only considerably reduces the number of training parameters but also generally produces consistencies greater than or equal to those of FT. This is a great advantage that perfectly matches the objective of applying PI.

| Model | | KB-BoolQ $\mathcal{A}_{test}\uparrow$ | KB-COPA $\mathcal{A}_{test}\uparrow$ | KB-WiC $\mathcal{A}_{test}\uparrow$ | KB-HellaSwag $\mathcal{A}_{test}\uparrow$ | KB-SentiNeg $\tau_{neg}\downarrow$ |
|---|---|---|---|---|---|---|
| KoRoB$_B$ | FT (111M) P-only | 78.2 | 60.4 | 79.1 | **78.2** | 8.5 |
| | P+C | **78.7** | 65.3* | 81.3* | 78.1 | **7.4*** |
| | PI (1.2M) P-only | 77.2 | 74.1 | 79.4 | 77.1 | 7.1 |
| | P+C | **78.4** | 75.0† | 80.7* | **77.4** | **6.5*** |
| KoRoB$_L$ | FT (338M) P-only | 86.5 | 75.7 | 85.1 | 83.4 | 5.5 |
| | P+C | **86.9** | 82.5* | 86.3* | **83.6** | 4.6* |
| | PI (2.6M) P-only | 86.7 | 83.2 | 85.5 | 84.8 | 5.5 |
| | P+C | 87.6* | 83.9† | **85.7** | 85.1 | **4.5*** |

Table 4: Average KoBEST performance of KoRoBERTa-base (KoRoB$_B$) and KoRoBERTa-large (KoRoB$_L$). $\mathcal{A}_{test}$ and $\tau_{neg}$ represent the accuracy and negational consistency on the test set, respectively. The best performance is in bold. The figures of the proposed approach (P+C) show a significant difference compared to our counterpart (P-only) with $p$-value $< 0.05$ (*) and $p$-value $< 0.1$ (†) using the t-test.

**Comparison with baseline models.** We compared the performance of our models with three baseline approaches. The first model is Dict-BERT (Yu et al., 2022), a model that employs the definition of rare words from the dictionary during the pre-training stage. The second approach is paraphrased data augmentation (Sem-Aug), which is also known as adversarial training (Yoo and Qi, 2021), which additionally leverages paraphrased versions of the original data for training. The last approach is semantic consistency regularisation (Sem-CR) (Elazar et al., 2021), which trains a model to generate nearly identical predictive distributions for both the original and paraphrased data instances through a consistency regularisation term. Referring to prior studies (Elazar et al., 2021; Kumar and Joshi, 2022), we defined the loss function of Sem-CR approach as follows:

$$\mathcal{L} = \mathcal{L}_{ce} + \lambda JS(o||p), \tag{7}$$

where $\mathcal{L}_{ce}$ is the cross-entropy loss, and $JS$ refers to the Jensen-Shannon divergence. $o$ and $p$ are the original and corresponding paraphrased data points, respectively. Similarly to Elazar et al. (2021), we conducted tuning of parameters ($\lambda \in 0.1, 0.5, 1$) to pick the best model. To generate paraphrased data for Sem-Aug and Sem-CR, we employed the EMBEDDING method of TextAttack (Morris et al., 2020), which replaces a certain portion of words (15% in our experiment) with their neighbours in the counter-fitted embedding space. Sem-Aug and Sem-CR were applied to Roberta-large. To verify the impact of additional knowledge add-up, we incorporated the weights of a RoBERTa-large model trained on the meaning-matching task (Jang et al., 2022b). The model employed dictionary data in a distinct manner compared to our approach and showed an improved performance in understanding negation expressions and antonyms. We hypothesised that adding the parameters trained with a different inductive bias could further enhance consistencies, particularly in negational consistency in our experiment, while maintaining overall performance.

The results are summarised in Table 3. First, we observe that our approach outperforms Dict-BERT in terms of both accuracy and consistency by a large margin. As the performance gap could be attributed to differences in model size, we also compared the performance between DictBERT and RoB$_B$-PI-(P+C) in Table 2. Regarding the accuracy, the latter exhibited a higher performance than the former in all five downstream tasks, with statistical significance at a $p$-value $< 0.05$. When it comes to consistency, the latter produced better results across 9 out of 15 test cases, which was statistically significant with a $p$-value $< 0.05$, while the former outperformed the latter only in 2 test cases. The results demonstrate that our proposed approach can achieve a better inductive bias compared to DictBERT.

We ascertained that both Sem-Aug and Sem-CR produced lower semantic inconsistency compared to our approach in most test cases (3 out of 5 on average). However, they were substantially worse in other consistency types. These findings provide evidence supporting our claim that data augmentation and consistency regularisation are restricted in addressing a specific consistency type, whereas our approach can improve multiple consistency types concurrently. Furthermore, both methods exhibited a lower accuracy across all downstream tasks and encountered a complete failure in SNLI, which offsets their lower transitive inconsistency. We believe that a primary factor is the imperfect performance of the paraphrasing method, which can confuse

models during training. Several examples of such cases are presented in Table 7 in Appendix C. Finally, we confirmed that the P+C+M model exhibited a similar or better performance than the P+C model. Specifically, the former showed statistically significant improvements in accuracy for 2 out of 5 tasks and achieved a lower inconsistency in 5 out of 15 test cases. These improvements were predominantly observed in negational consistency (3 out of 4 test cases). No statistically significant differences were observed in other test cases. These results verify our aforementioned hypothesis regarding the effect of integrating additional parameters trained with distinct inductive bias.

## 3.3 Experiments on the Korean Dataset

As mentioned earlier, flexible applicability to other languages is one of the key advantages of our approach. Hence, we applied our method to the Korean language to verify this.

### 3.3.1 Training the Korean CRM Model

We collected 1.6M *word-definition* pairs from the Korean Dictionary made by the National Institute of Korean Language.[5] Regarding the backbone PLM, we used Korean RoBERTa models (Park et al., 2021). The same computing resources and training hyperparameter settings described in Section 3.1 were used for training.

### 3.3.2 Experiments on the KoBEST Dataset

We assessed the proposed approach on the KoBEST dataset (Kim et al., 2022), a recently proposed challenging benchmark similar to the Korean-GLUE benchmark (Park et al., 2021). The dataset consists of five downstream tasks: Boolean Question (KB-BoolQ), Choice of Plausible Alternative (KB-COPA), Words in Context (KB-WiC), HellaSwag (KB-HellaSwag), and Sentiment Negation (KB-SentiNeg). Each task is designed to evaluate the model's understanding of context information, causality, word's meaning, time flow, and negation expressions, respectively. The SentiNeg task is a sentiment analysis task, but the test set is composed of negated training sentences where adding the negation expressions changes the polarity. Therefore, it supports the evaluation of negational consistency. Detailed hyperparameter settings for each task are described in Appendix B.

Table 4 presents the experimental results. Overall, the results exhibit the analogous trend observed

in English experiments. First, exploiting the CRM weights positively influences both fine-tuning and parameter integration cases, recording statistically significant performance improvements in 3 out of 5 tasks on average. A marginal difference is that for a large-size model, i.e., RoBERTa-large, using CRM shows an almost similar performance to using PLM-only in the GLUE benchmark experiments, while the former outperforms the latter in the KoBEST experiments. We presume that the leading causes are that (1) we collected more data (about four times higher) for training Korean CRM than the English counterpart, and (2) KoBEST downstream tasks require a higher language understanding ability, where the effects of CRM can be more pronounced. Second, although we examined only one test case due to the nonexistence of the Korean dataset for evaluating multiple consistency types, the negational consistency is significantly improved after applying CRM, which is in line with our results on English datasets. It would be interesting future work to expand the BECEL dataset to Korean and investigate the effect of CRM. The experimental results support that our proposed approach is easily applicable and works well in other languages beyond English.

## 4 Related Work

**Analysis on consistency.** Most widely conducted studies regarding consistency are based on *semantic equivalence*, i.e., a model should generate the same predictions for text inputs conveying identical meanings. Several works observed that when PLMs perform masked language modelling, they generate different predictions for queries where an object is replaced with its plural form (Ravichander et al., 2020) or for paraphrased queries (Elazar et al., 2021). Elazar et al. (2021) introduced a consistency regularisation method to alleviate the inconsistent issue that forces PLM to produce similar predictive distributions for the original and paraphrased queries. Zhou et al. (2022) proposed a novel consistency regularisation technique known as "prompt consistency", designed for zero-shot learning. This method encourages consistent predictions across various sets of prompts. Another line of work investigated PLMs' violation of logical properties, such as symmetry (Wang et al., 2019c; Li et al., 2019; Kumar and Joshi, 2022) and transitivity (Li et al., 2019; Asai and Hajishirzi, 2020; Lin and Ng, 2022). Recently, Jang et al. (2022a) proposed the definition of language

---

[5]https://github.com/spellcheck-ko/korean-dict-nikl

model's consistency based on behavioural consistency and classified previous works into three large categories: semantic, logical, and factual consistency. Semantic consistency contains works based on *semantic equivalence*, which should always be satisfied regardless of the tasks and data. Logical consistency only applies to tasks where a certain property holds and consists of four subcategories: negational, symmetric, transitivity, and additive consistency. Unlike the abovementioned studies that concentrated on the prediction inconsistency of language models (LMs), several research delved into the inconsistency of explanations generated by PLMs and large language models (LLMs) (Jung et al., 2022; Wang et al., 2023; Jang et al., 2023).

**Adaptation technique.** Adapter modules are invented to efficiently fine-tune PLMs by maintaining pre-trained weights fixed and introducing only a few trainable parameters. This is especially efficient when training multiple downstream tasks, as it allows training a few new parameters per task without re-updating previously trained weights. Many prior works inserted a low-rank residual adapter between layers (Rebuffi et al., 2017; Houlsby et al., 2019; Lin et al., 2020). Karimi Mahabadi et al. (2021) proposed the COMPACTER method, which exploits the Kronecker product for a low-rank parameterisation, unlike the previous approaches that use down- and up-projections. Recently, Hu et al. (2022) proposed an approach named LoRA. A distinctive difference between LoRA and the aforementioned methods is that the learnable weights of the former method are aggregated with pre-trained weights during the inference phase and, hence, do not increase the latency.

## 5 Summary and Outlook

Trustworthiness is a highly recommended property that a good natural language processing (NLP) model should satisfy, especially in risk-sensitive fields, such as the legal or medical domain. However, recent studies have revealed that modern PLMs make so many mistakes that humans rarely commit and, hence, are not perfectly reliable. One such faulty phenomenon is PLMs' inconsistent behaviours. Many studies proposed a remedy to resolve the issue by using data augmentation and designing loss functions that penalise inconsistent actions. However, such methods have disadvantages in that they can only deal with specific consistency types, hardly apply to low-resource languages, and require expensive training resources for large PLMs.

In this paper, we proposed an approach that improves overall consistencies in a practical manner. We hypothesised that a compelling reason for inconsistent behaviours is the lack of meaning-understanding ability and, hence, attempted to enhance PLMs' meaning awareness based on the conceptual role theory. We designed a learning task that leverages *word-definition* pairs in a dictionary to capture more precise interconnection between concepts and named the learned model CRM. Next, we employed a parameter integration technique that combines the weights of CRM and PLM to maximally exploit previously- and newly-learned knowledge. Our findings suggest that improved meaning awareness largely contributes to enhancing multiple consistency types concurrently. In addition, we observed that further improvements are achievable through incorporating additional parameters possessing distinct inductive bias. Finally, we verified that our approach can readily apply to languages beyond English. Our experimental results suggest that investigating human cognition and transplanting it to learning objectives could be a key to trustworthy language AI.

## Limitations

For Korean dataset experiments, we were only able to measure negational consistency due to the lack of a publicly available consistency evaluation dataset in Korean. In our English experiments, we collected 455K word-definition pairs to train CRM. Despite our attempts to collect more data by leveraging the Oxford Dictionary API [6], we were unable to obtain approval for the API key. Considering the experimental results in Korean experiments, where CRM was trained with 1.6M word-definition pairs, we could anticipate further performance improvements in the English experiments with more word-definition pairs.

## Acknowledgements

This work was partially supported by the Alan Turing Institute under the EPSRC grant EP/N510129/1 and by the AXA Research Fund. We also acknowledge the use of Oxford's ARC facility, of the EPSRC-funded Tier 2 facility JADE II (EP/

---

[6]https://developer.oxforddictionaries.com/

T022205/1), and of GPU computing support by Scan Computers International Ltd.

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

## A  PLMs and Distributional Hypothesis

Sinha et al. (2021a) showed in four steps that training BERT (Devlin et al., 2019) with the masked language modelling objective is not much different from training word2vec (Mikolov et al., 2013), a representative method based on the distributional hypothesis. We briefly recall the material here.

We recall the parameterisation of skip-gram word vectors (Mikolov et al., 2013):

$$p(t|w;\theta) = \frac{e^{f(t,w)}}{\sum_{t' \in V} e^{f(t',w)}} \, ,$$

where $f$ is a dot product, $V$ is the set of all possible words, and $t$ and $w$ are vectors of the target word and a word in the context, respectively. During training, negative sampling is used within a window size, and the loss function, which is $\log \sigma(w \cdot t) + k \cdot \mathbb{E}_{t' \in P} \log \sigma(-w \cdot t')$, is optimised over the context $C(w_i) = \{w_{i-k}, ..., w_{i-1}, w_{i+1}, ..., w_{i+k}\}$ for a word index $i$, a window size of $2k$, and unigram probability distribution $P$.

**Step 1: BPE:**  Computing the full softmax probability entails a huge matrix multiplication with a large vocabulary $V$. Instead, BERT exploits subword units that guarantee a smaller total vocabulary $U$ in the softmax denominator.

**Step 2: Defenestration:**  Next, instead of using the local context window, BERT uses the entire sentence while making the target word: $C(t) = \{w \in S : w \neq t\}$, where $S$ is the sentence including the word $w$.

**Step 3: Non-linearity:**  The pairwise word-level dot product $f(w, t)$ is replaced with a non-linear function. We can imagine using a sequence of multi-head self-attention layers $g(t, C(t))$ in BERT. Now, we get the parameterisation of BERT as follows:

$$p(t|C(t);\theta) = \frac{e^{g(t,C(t))}}{\sum_{t' \in U} e^{g(t',C(t))}} \, .$$

**Step 4: Sprinkle data and compute:**  Now, the model $g$ is trained with tremendous amounts of documents. After the pre-training, the parameters of the model $g$ are also updated during the fine-tuning on downstream tasks.

The same method can be applied to any PLMs trained with a masked language modelling objective. We can also show in a similar way that generative PLMs trained with an auto-regressive language modelling objective, e.g., GPT-2 (Radford et al., 2019) and -3 (Brown et al., 2020), are also based on the distributional hypothesis by simply changing $C(t)$ in **Step 2** as follows:

$$C(t) = \{w_0, w_1, ..., w_{t-1}\} \, ,$$

where $w_{t-1}$ denotes a word that appears right before the word $t$.

## B  Hyperparameters

This section presents the hyperparameter values used for the downstream task experiments.

### B.1  Experiments on English datasets

For the GLUE (Wang et al., 2019b) and BECEL datasets (Jang et al., 2022a), we used the same hyperparameter settings. We set the maximum sequence length to 256, trained all models for 15 epochs, and chose the model with the best validation accuracy. The batch size was set to 32. All models are trained with the AdamW optimiser (Loshchilov and Hutter, 2017) with a linear learning rate scheduler decaying from $1e^{-2}$ and warm-up ratio of 0.06. The learning rate was set to $2e^{-5}$ for the FT models and $5e^{-4}$ for the PI models, respectively.

### B.2  Experiment on the KoBEST dataset

Table 5 shows the hyperparameter values used for training the PI models on KoBEST downstream tasks. For training the FT models, small changes are made for the learning rate, $1e^{-5}$ for the KB-WiC and KB-HellaSwag, and $5e^{-6}$ for the others.

| | BoolQ | COPA | WiC | HellaSwag | SentiNeg |
|---|---|---|---|---|---|
| max-len | 256 | 128 | 128 | 176 | 128 |
| epochs | 10 | 10 | 10 | 10 | 10 |
| b-size | 16 | 16 | 16 | 8 | 16 |
| lr | $5e^{-4}$ | $5e^{-4}$ | $5e^{-4}$ | $5e^{-4}$ | $5e^{-4}$ |
| weight decay | 0.1 | 0.1 | 0.1 | 0.1 | 0.1 |

Table 5: Hyperparameters used for training the PI models on KoBEST downstream tasks.

# C   Examples

| Raw Dictionary Data | |
|---|---|
| Word | Definition |
| *happy* | 1) feeling or showing pleasure; pleased |
| | 2) giving or causing pleasure |
| *unhappy* | 1) not happy; sad |
| | 2) not pleased or satisfied with somebody |
| | or something |
| **Training Instances** | |
| happy feeling or showing pleasure; pleased giving or causing pleasure | |
| unhappy not happy; sad not pleased or satisfied with somebody or something | |

Table 6: Examples of word-definition pairs in a dictionary and their concatenated version for training the CRM.

| PREMISE: An older man is drinking orange juice at a restaurant. | |
|---|---|
| ORIGINAL HYPOTHESIS: A man is drinking juice. | PARAPHRASED HYPOTHESIS: A man is alcohol juice. |
| LABEL: Entailment | LABEL: Entailment |
| PREMISE: A person on skis on a rail at night. | |
| ORIGINAL HYPOTHESIS: They are fantastic skiiers. | PARAPHRASED HYPOTHESIS: They are terrific skiiers. |
| LABEL: Neutral | LABEL: Neutral |
| PREMISE: High fashion ladies wait outside a tram beside a crowd of people in the city. | |
| ORIGINAL HYPOTHESIS: Women are waiting by a tram. | PARAPHRASED HYPOTHESIS: Women are waiting by a streetcar. |
| LABEL: Entailment | LABEL: Entailment |

Table 7: Examples of imperfect paraphrased hypotheses of the SNLI task generated by TextAttack.