# OpenReview forum: "Improving Language Models’ Meaning Understanding and Consistency by Learning Conceptual Roles from Dictionary"
_EMNLP/2023/Conference — EMNLP 2023 Main_

### Official Review · Reviewer_zTcT · 2023-08-04

**Soundness:** 4

**Excitement:**

4: Strong: This paper deepens the understanding of some phenomenon or lowers the barriers to an existing research direction.

**Paper Topic And Main Contributions:**

To address the inconsistent issues of PLMs, this paper introduces a learning framework that leverages word-definition pairs in a dictionary to capture the interconnection between concepts to enhance the capability of meaning-understanding for PLMs. The authors further integrate learned parameters (from multiple knowledge sources) by an aggregation method to improve performance and reduce inconsistency. Especially the proposed approach can concurrently enhance multiple types of consistency simultaneously. Experimental results demonstrate the effectiveness of both English and Korean Language datasets.

**Questions For The Authors:**

Q1: Lines #252-#255: Where are these previous studies from? Please cite them.

Q2: Lines #296-297 There are 455K collected word-definition pairs for training the CRM model. Will the performance be improved if the training pair increases? It is suggested to make a comparison across different training size scales.

Q3: Table 3: How to integrate the P+C+M model? By Eq(6)?

Q4: Lines #539-#540. Did the author try multilingual Roberta PLM (instead of Korean RoBERTa)? If you did, are there differences in the results between them?

Q5: Lines #667-#670: If there is a problem with the collection of dictionaries, how about using the synset strings (from WordNet) as training samples?


**Reasons To Accept:**

This paper proposed a simple but effective learning framework to enhance the meaning-understand capability of PLMs. The authors conducted experiments on two/one English/Korean public datasets, and the results supported the claims and proved the effectiveness. Furthermore, the proposed parameter integration approach (by aggregating additional weights learned from various resources) not only integrates knowledge from different sources to enhance the understanding capability of PLMs but also avoids the risk of catastrophic forgetting. The related experiments, findings, and observations will benefit further research.

**Reasons To Reject:**

According to the experimental results, the model size and training approach of the PLM will affect the improved performance of the proposed method. The authors use RoBERTa as the backbone model. This work does not discuss whether other PLMs (such as BERT) will have similar results. Furthermore, large language models' (LLM) training parameters (i.e., the stored knowledge) are significantly larger than those of PLMs. There is no relevant experiment and discussion on whether the inconsistency issues in PLM also occur in LLM in this work.

**Reproducibility:**

3: Could reproduce the results with some difficulty. The settings of parameters are underspecified or subjectively determined; the training/evaluation data are not widely available.

**Reviewer Confidence:**

3: Pretty sure, but there's a chance I missed something. Although I have a good feel for this area in general, I did not carefully check the paper's details, e.g., the math, experimental design, or novelty.

---

> ### Author Rebuttal · Authors · 2023-08-29
>
> Thank you for your constructive review. Please find our answers to your questions below:
>
> Answer to Q1: The prior studies refer to the work of Hu et al. (2022) for parameter integration and the RoBERTa paper (Liu et al., 2019). We will mention these studies in lines #252-#255.
>
> Answer to Q2: We do believe that more training pairs can improve the performance, because (1) it is a generally observed phenomenon in the NLP field that more data increase the performance, and (2) the experiments on the Korean dataset, which employed more training pairs, demonstrate a larger performance gain. We will conduct experiments with smaller training pairs and add the results to the final manuscript.
>
> Answer to Q3:  The weights of P, C, and M are integrated through the averaging aggregation method. We will add more detailed explanations and update the equation to avoid possible confusion.
>
> Answer to Q4: We did not compare the performance of the Multilingual RoBERTa model, as our primary intent for the Korean dataset experiments was to ascertain our approach’s applicability to low-resource languages. The usage of multilingual models can prevent precise evaluation, because such models can benefit from the extensive datasets in foreign languages. Hence, to minimise the influence stemming from external factors, multilingual models are not considered. However, we agree that it would be interesting to see how the consistency performance would be affected by the usage of multilingual models. We will conduct additional experiments regarding this.
>
> Answer to Q5: We tried to also use WordNet synsets but observed that the words were highly overlapping with the words that we collected. Hence, we did not use the WordNet dataset for our experiments.

---

### Official Review · Reviewer_DNHZ · 2023-08-05

**Soundness:** 4

**Excitement:**

4: Strong: This paper deepens the understanding of some phenomenon or lowers the barriers to an existing research direction.

**Paper Topic And Main Contributions:**

The paper aims to develop more robust and consistent LMs by incorporating word definition knowledge into LMs by leveraging external dictionary data. Current LMs have inconsistent behaviour like different predictions for semantically identical text or displaying logical contradictions. This work hypothesizes that current pretraining mechanisms rely on co-occurance contexts which is insufficient and that using word definitions provides a more reliable ‘meaning’ based context which is more reliable. They train an auxiliary model (CRM) using dictionary data and propose parameter integration methods to more efficiently incorporate the model with existing one. Their experiments show that the model has comparable or better NLU performance on GLUE and better performance on consistency settings like BECEL. Additionally, they aslo show that this model/technique can be easily extended beyond English by showing similar experiments on Korean datasets.

The main contributions of this paper are:

1. Leveraging dictionary data to improve robustness and consistency

2. Parameter efficient training and methods to incorporate new knowledge with existing models

3. Method applicable to new languages with readily available dictionary data.

**Questions For The Authors:**

A. How was CRM model trained ? Did you use MLM for this model as well ? How many steps was the model trained for ?
B. Line 399 - which figures ?

**Reasons To Accept:**

1. Interesting idea to leverage easily available dictionary data for improved contextual information during pre-training
2. Parameter efficient model can enable improving the model in a resource efficient manner.
3. Generalizable method - helps improve multiple consistency types with the same training setup. Additionally easily extendable to new languages also - dictionary data is easily available in most languages.

**Reasons To Reject:**

1. Missing baselines - some important baselines are missing which are necessary for a complete picture of the model. These suggested experiments are not parameter efficient, but will provide the trade-off details wrt performance.

   i) C-only model - how well does CRM model do the task is PI necessary ?

   ii) Continued training on P instead of PI - again to ablate whether PI is necessary, continuing to train the original on the original corpus for same steps and additionally on the dictionary corpus for same steps will help understand benefits of current weight integration setup.


2. Experiment and Evaluation details are a bit hard to follow - there are many decision choices on data (some datasets included but others aren’t) and in evaluation (modifying evaluation metric) which are mentioned but in a high-level manner. Without giving details, it’s hard to understand why certain experimental setups were chosen and hard to even reproduce using details in the paper.

i) 354-360 Can you give more details on why additive consistency was not reported. Yes PLMs perform well, but does this model perform well as well ?

ii) Line 388 - Why was evaluation metric modified (exclusively considers instances in which the model makes accurate predictions) - why is including inaccurate predictions problematic ? maybe more details on misestimation might help ?

**Reproducibility:**

3: Could reproduce the results with some difficulty. The settings of parameters are underspecified or subjectively determined; the training/evaluation data are not widely available.

**Reviewer Confidence:**

3: Pretty sure, but there's a chance I missed something. Although I have a good feel for this area in general, I did not carefully check the paper's details, e.g., the math, experimental design, or novelty.

---

> ### Author Rebuttal · Authors · 2023-08-29
>
> Thank you for your constructive review. Please find our answers to your comments and questions below.
>
> Answer to 1.i)+ii): We tested the C-only model and observed that it showed a better consistency than P-only, but the improvements were less significant than P+C. We will add this to the manuscript. When it comes to the second baseline, we did not test the approach, because continued training on the dictionary data can cause catastrophic forgetting of the original corpus, which can overestimate the advantage of the CRM parameter integration. However, we agree that it would be more helpful to understand the benefits of the parameter integration, if our concerns are verified with the performance of the second baseline. We will perform the experiments and add them to the final manuscript.
>
> Answer to 2.i) : The additive consistency was not reported, mainly because previous experiments in BECEL paper showed that every PLM is highly consistent on the additive consistency. We also ascertained through pilot tests that the models that employed the CRM model (both FT and PI) were also highly consistent on the additive consistency, so it is excluded from the experimental scope of the paper. We will mention this in the final manuscript.
>
> Answer to 2.ii): The inclusion of inaccurate predictions can prevent precise evaluations, because logical consistencies, such as negation consistency and symmetric consistency, are conditional consistency types. For example, negation consistency applies when the label is ``Equivalent'' for the STS task. Consider the below examples of the MRPC task:
>
> S1: In the evening, he asked for six pepperoni pizzas and two six-packs of soft drinks, which officers delivered.
>
> S2: In the evening, he asked for six pizzas and soda, which police delivered.
>
> S2-neg: In the evening, he asked for six pizzas and soda, which police did not deliver.
>
> The gold label of the S1-S2 pair is “Equivalent”, and provided the model generates a correct answer for the S1-S2 pair, predicting the relation between S1-S2-neg as “Equivalent” is a violation of negation consistency. However, assume that the model believes that the answer of the S1-S2 pair is ``Not Equivalent''. In this case, according to the original metric, it is considered as the violation of negation consistency if the model generates “Not Equivalent” as the answer of the S1-S2-neg pair despite it is the correct answer. On the other hand, if the model generates “Equivalent” as the answer of the S1-S2 pair, it is considered as a consistent prediction despite it is a wrong answer. This example shows that the condition of logical consistencies should be applied based on the model’s belief not the gold-label of data instances. Therefore, we modified the metrics to consider only the instances where the models make correct predictions. We will add this to the final manuscript with an additional page.
>
> Answer to "How was CRM model trained?": The CRM model is trained by using the MLM objective. The base and large models are trained for 500K and 700K steps, respectively, as outlined in lines #303-304.
>
> Answer to "which figures?": The word ‘figure’ denotes the number of test cases where the RoBERTa-large model showed statistically significant improvements in FT and PI. We will change the terminology to avoid confusion.

---

### Official Review · Reviewer_37Br · 2023-08-05

**Soundness:** 3

**Excitement:**

2: Mediocre: This paper makes marginal contributions (vs non-contemporaneous work), so I would rather not see it in the conference.

**Paper Topic And Main Contributions:**

Core idea: This paper addresses the problem of  inconsistent behavior by improving “PLMs’ meaning awareness”. They state that existing methods addressing inconsistency  i) require expensive resources and ii) are only able to address a specific type of consistency and are unable to handle others. They hypothesis that insufficient meaning awareness is the root cause of inconsistency.

Method:They continue training a PLM (Roberta) on a corpus consisting of a collection of word-definition pairs (455k) which they call conceptual role model (CRM). They then fine-tune an efficient integration of the parameters of the original PLMs and CRM on downstream tasks.

Evaluation: They evaluate on English GLUE and BECEL (a consistency dataset) and Korean KoBEST (similar to Korean-GLUE). They compare to English/Korean Roberta model and baseline methods targeting consistency.

**Questions For The Authors:**

- Maybe I misunderstood your ablations but did you also train i) Roberta alone on your data (so P-only but fine-tuned on your data) and ii) continued training Roberta for equally many steps on Wikipedia data? The first would be a sanity check for the more elaborate architecture you are using and the second would be a sanity check that it is not simply more data that improves performance but it is actually the type of data you are adding that is better than other data.

**Reasons To Accept:**

- Performance on the consistency dataset BECEL is better than their Roberta baseline model.
- Performance on GLUE is on par with Roberta as is, meaning that their method does not degrade general performance.

**Reasons To Reject:**

- They test their method on Roberta only. I doubt that their method is generally useful specifically in the context of more recent larger models where these definitions seem like a drop in the ocean.
- There is related work fusing additional knowledge into PLMs. I wonder how their method's performance differs from other forms of continued training. (See also my question.)
- The authors formulate conclusions which are not backed by their experimental findings. In section 3.2 their results are not showing systematic improvements over GLUE. The only thing one can take away from their GLUE results is that it does not degrade general performance. They could have presented it as that but instead they conclude that section with "The experimental results suggest that CRM weights provide more abundant linguistic knowledge, which can help enhance the understanding of text meaning.".

**Reproducibility:**

5: Could easily reproduce the results.

**Reviewer Confidence:**

3: Pretty sure, but there's a chance I missed something. Although I have a good feel for this area in general, I did not carefully check the paper's details, e.g., the math, experimental design, or novelty.

**Typos Grammar Style And Presentation Improvements:**

There are quite a number of places where you made broader references/statements which were very vague. E.g., 'The non-humanlike behaviour of contemporary pre-trained language models (PLMs) is a leading cause undermining their trustworthiness.'

---

> ### Author Rebuttal · Authors · 2023-08-29
>
> Thank you for your review. Please find our answers to your questions below:
>
> A1: By "related work fusing additional knowledge into PLMs", you mean integrating knowledge bases with PLMs? If so, note that these approaches are not our proper baselines. The term “knowledge add-up” that we used implies integrating parameters of multiple models having different inductive biases to improve the understanding of textual meaning, not fusing knowledge bases with PLMs. We will choose an alternative terminology to avoid confusion.
>
>
> A2: Note that, in Section 3.2, we are not arguing that the introduction of CRM achieves a higher textual meaning understanding based on the GLUE results. Instead, we claim that the CRM model contains more linguistic knowledge than the original PLM. As illustrated in lines #337-343, the experimental results show that the employment of the CRM model generates an improved performance in the COLA task, which is designed to check grammatical errors and requires linguistic knowledge. Also, the improvements were more significant in RoBERTa-base that stores less pre-trained knowledge. Based on these results, we demonstrated that “The experimental results suggest that CRM weights provide more abundant linguistic knowledge”.
>
> When it comes to the rest of the statement in lines #345-346, we are not arguing at this stage that our proposed approach achieved a better textual meaning understanding based on the GLUE results. Our intention is that the abundant linguistic knowledge provided by CRM weights might help the model to improve the understanding of textual meaning and, thereby, enhance consistency. We will state this more clearly to avoid confusion.
>
>
> A3: By “P-only but fine-tuned on our data”, you mean the CRM-only model? We tested this model and observed that the consistency is generally improved than P-only, but the improvements were less significant than P+C approaches. We will add this to the final manuscript to provide more comprehensive information. Regarding the second ablation model that you mentioned, we did not perform experiments, but we do believe that the result can highlight the importance of data type. We will conduct this experiment and add the results to the final manuscript.
>
> A4: As for what you write under “Typos, Grammar, Style, and Presentation Improvement”, we did not include references for the statement that you mentioned, because it is demonstrated in the abstract, but we did add references regarding that statement in lines #33-34. Could you let us know the other vague references/statements that we made? We will fix this in our final manuscript.

---

### Meta-Review · Area_Chair_vevw · 2023-09-19

**Recommendation:** 4

**Metareview:**

The paper presents a method to improve the consistency of pretrained LMs predictions by leveraging structured semantic information extracted from a dictionary. The augmented model is able to outperform baseline ones on the GLUE and BECEL datasets. Though the method is not particularly innovative, there is consensus about the methodological soundness of the paper, which makes it a valuable contribution. A possible limitation is the fact that the method is only applied to RoBERTa, lacking a larger experimentation with other LMs. However, this does not negatively affect the general strength of the paper.

---

### Decision · Program_Chairs · 2023-10-07

**Decision:**

Accept-Main

**Comment:**

The paper presents a method to improve the consistency of pretrained LMs predictions by leveraging structured semantic information extracted from a dictionary. The augmented model is able to outperform baseline ones on the GLUE and BECEL datasets. Though the method is not particularly innovative, there is consensus about the methodological soundness of the paper, which makes it a valuable contribution. A possible limitation is the fact that the method is only applied to RoBERTa, lacking a larger experimentation with other LMs. However, this does not negatively affect the general strength of the paper.